# Learning Disentangled Representations for Perceptual Point Cloud Quality Assessment via Mutual Information Minimization

**Ziyu Shan**[*], **Yujie Zhang**[*], **Yipeng Liu**, **Yiling Xu**[†]
Cooperative Medianet Innovation Center, Shanghai Jiao Tong University
{shanziyu, yujie19981026, liuyipeng, yl.xu}@sjtu.edu.cn

## Abstract

No-Reference Point Cloud Quality Assessment (NR-PCQA) aims to objectively assess the human perceptual quality of point clouds without relying on pristine-quality point clouds for reference. It is becoming increasingly significant with the rapid advancement of immersive media applications such as virtual reality (VR) and augmented reality (AR). However, current NR-PCQA models attempt to indiscriminately learn point cloud content and distortion representations within a single network, overlooking their distinct contributions to quality information. To address this issue, we propose DisPA, a novel disentangled representation learning framework for NR-PCQA. The framework trains a dual-branch disentanglement network to minimize mutual information (MI) between representations of point cloud content and distortion. Specifically, to fully disentangle representations, the two branches adopt different philosophies: the content-aware encoder is pretrained by a masked auto-encoding strategy, which can allow the encoder to capture semantic information from rendered images of distorted point clouds; the distortion-aware encoder takes a mini-patch map as input, which forces the encoder to focus on low-level distortion patterns. Furthermore, we utilize an MI estimator to estimate the tight upper bound of the actual MI and further minimize it to achieve explicit representation disentanglement. Extensive experimental results demonstrate that DisPA outperforms state-of-the-art methods on multiple PCQA datasets.

## 1 Introduction

With recent advances in 3D capture devices, point clouds have become a prominent media format to represent 3D visual content in various immersive applications, such as autonomous driving and virtual reality [4, 43]. These extensive applications stem from the rich information provided by point clouds (*e.g.*, geometric coordinates, color). Nevertheless, before reaching the user-client, point clouds inevitably undergo various distortions at multiple stages, including acquisition, compression, transmission and rendering, leading to undesired perceptual quality degradation. Accordingly, it is necessary to develop an effective metric that introduces human perception into the research of point cloud quality assessment (PCQA), especially in the common no-reference (NR) situation where pristine reference point clouds are unavailable.

In recent years, many deep learning-based NR-PCQA methods [21, 47, 51, 32, 3] have shown remarkable performance on multiple benchmarks, which can be applied directly to 3D point cloud data or 2D rendered images. Most of these methods [47, 22, 51, 3] tend to learn a unified representation for quality prediction, ignoring the fact that perceptual quality is determined by both point cloud

---

[*]Ziyu Shan and Yujie Zhang contribute equally to this work.
[†]Corresponding author

38th Conference on Neural Information Processing Systems (NeurIPS 2024).

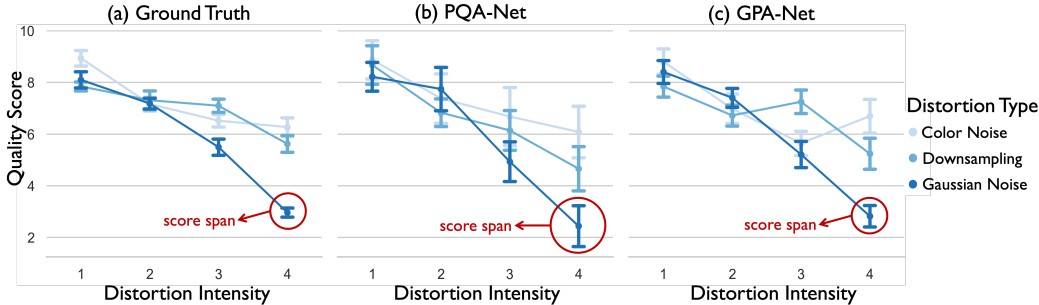

Figure 1: Statistics of SJTU-PCQA (part) [46] and predicted quality scores of NR-PCQA models (PQA-Net [21] and GPA-Net [32]). Quality scores of different distortion types are in lines of different colors. Red circles are to highlight the score span of different contents with the same distortion.

content information and distortion pattern. Although some other models [21, 33] alternately learn content and distortion representations through different training objectives, they are still based on a single-branch network and thus may lead to highly entangled features in the representation space.

From the perspective of visual rules, insufficient disentanglement between representations of point cloud content and distortion disobeys the perception mechanisms of the human vision system (HVS), further limiting performance improvement. In fact, many studies [35, 16] highlight the distinct visual processing of high-level (*e.g.*, semantics) and low-level information (*e.g.*, distortions) in different areas of the brain. Concretely, the left and right hemispheres of the brain are specialized in processing high-level and low-level information, respectively. These findings suggest a relatively disentangled processing mechanism in our brain, challenging existing methods that seek to learn these conflicting representations using a single network indiscriminately.

The difficulty of disentangled feature learning is relatively great for NR-PCQA due to data imbalance. Specifically, although a wide range of distortion types and intensities in current PCQA datasets can enable the learning of robust low-level distortion representations, it is non-trivial to learn the representations of point cloud content that lies in a considerable high dimensional space because these PCQA datasets are extremely limited in terms of content (*e.g.*, up to 104 contents in LS-PCQA [23]). This data limitation can lead to overfitting of NR-PCQA models regarding point cloud content, that is, when the content changes, the prediction score changes in the undesired manner, even with the same distortion pattern. As illustrated in Figure 1 (b) and (c), the NR-PCQA models PQA-Net [21] and GPA-Net [32] correctly predict the trend of quality degradation with increasing distortion intensity, but their predicted score spans deviate a lot from the ground truth in Figure 1 (a), where the content varies but the distortion pattern remains intact. Based on these observations, we expect a new disentangled representation learning framework that can obey the separate information processing mechanism of HVS, and alleviate the difficulty of content and distortion representation learning introduced by data imbalance.

In this paper, we propose a new **Dis**entangled representation learning framework tailored for NR-**PCQA**, named DisPA. Motivated by the HVS perception mechanism, DisPA employs a dual-branch structure to learn representations of point cloud content and distortion (called content-aware and distortion-aware branches). DisPA has three steps to achieve disentanglement: 1) To address the problem introduced by data imbalance, we pretrain a content-aware encoder based on masked autoencoding strategy. Specifically, in this pretraining process, the distorted point cloud is rendered into multi-view images whose patches will be partially masked. The partially masked images are then fed into the content-aware encoder to reconstruct the rendered images of the corresponding reference point cloud. 2) To facilitate learning of distortion-aware representations, we decompose the distorted multi-view images into a mini-patch map through grid mini-patch sampling [40], which can prominently present local distortions and forces the distortion-aware encoder to ignore the global content. 3) Inspired by the utilization of mutual information (MI) in disentangled representation learning [8], we propose an MI-based regularization to explicitly disentangle the latent representations. Compared to simple linear correlation coefficients (*e.g.*, cosine similarity), mutual information can capture the nonlinear statistical dependence between representations [15]. To achieve this, we utilize an MI estimator to estimate a tight upper bound of the MI and further minimize it to achieve straightforward disentanglement. We summarize the main contributions as follows:

- We propose a novel disentangled representation learning method for NR-PCQA called DisPA, which obeys the particular HVS perception mechanism. To the best of our knowledge, DisPA is the first framework to explore representation disentanglement in PCQA.

- We propose the key MI-based regularization that can explicitly disentangle the representations of point cloud content and distortion through MI minimization.

- We conduct comprehensive experiments on three datasets (SJTU-PCQA [46], WPC [20], LS-PCQA [23]), and achieve superior performance over the state-of-the-art methods on all of these datasets.

## 2 Related Work

**No-Reference Point Cloud Quality Assessment.**    NR-PCQA aims to evaluate the perceptual quality of distorted point clouds without available references. According to the modalities, the NR-PCQA methods can be categorized into three types: projection-based, point-based and multi-modal methods. For the projection-based methods, various learning-based networks [21, 47, 52, 33, 34] adopt multi-view projection for feature extraction, while Zhang *et al.*[53] integrates the projected images into a video to conveniently utilize video quality assessment methods to evaluate the perceptual quality. Xie *et al.*[45] first computes four types of projected images (*i.e.*, texture, normal, depth and roughness) and fuses their latent features using a graph-based network. For the point-based methods, Zhang *et al.*[50] extracts carefully designed hand-crafted features, while Liu *et al.*[23] transforms point clouds into voxels and utilizes 3D sparse convolution to learn the quality representations. Some 3D native methods [37, 32, 39] divide point clouds into local patches and utilize hierarchical networks structurally like PointNet++ [29] to learn the representations. For the multi-modal methods, Zhang *et al.*[51] utilizes individual 2D and 3D encoders to separately extract features, and fuse them using a symmetric attention module. Other works [38, 3, 22] leverage various cross-modal interaction mechanisms to enhance the fusion between 2D and 3D modalities. Compared to previous methods that learn quality representations indiscriminately, our work solves quality representation disentanglement from a more essential perspective of mutual information, which reveals the intrinsic correlations between point cloud content and distortion pattern.

**Representation Learning for Image/Video Quality Assessment.**    As for image quality assessment (IQA), CONTRIQUE [25] learns distortion-related information on images with synthetic and realistic distortions based on contrastive learning. Re-IQA [31] trains two separate encoders to learn high-level content and low-level image quality features through an improved contrastive paradigm. QPT [54] also learns quality-aware representations through contrastive learning, where the patches from the same image are treated as positive samples, while the negative sample are categorized into content-wise and distortion-wise samples to contribute distinctly to the contrastive loss. QPTv2 [44] is based on masked image modeling (MIM), which learns both quality-aware and aesthetics-aware representations through performing the MIM that considers degradation patterns.

As for VQA, CSPT [5] learns useful feature representation by using distorted video samples not only to formulate content-aware distorted instance contrasting but also to constitute an extra self-supervision signal for the distortion prediction task. DisCoVQA [41] models both temporal distortions and content-related temporal quality attention via transformer-based architectures. Ada-DQA [19] considers video distribution diversity and employ diverse pretrained models to benefit quality representation. DOVER [42] divides and conquers aesthetic-related and technical-related (distortion-related) perspectives in videos, introduces inductive biases for each perspective, including specific inputs, regularization strategies, and pretraining. However, there is no current work to utilize mutual information (MI) to achieve representation disentanglement, which has not been explored in IQA/VQA.

**Mutual Information Estimation.**    Mutual information (MI) has been widely used as regularizers or objectives to constrain independence between variables [2, 7, 13, 14]. Hjelm *et al.*[14] performs unsupervised representation learning by maximizing MI between the input and output of a deep neural network. Kim *et al.*[17] learns disentangled representations by encouraging the distribution of representations to be factorial and hence independent across the dimensions. Moreover, MI minimization has been drawing increasing attention in disentangled representation learning [6, 15, 55]. Chen *et al.*[8] introduces a contrastive log-ratio upper bound for mutual information estimation, and extends the estimator to a variational version for general scenarios when only samples of the

joint distribution are obtainable. Dunion *et al.*[11] minimizes the conditional mutual information between representations to improve generalization abilities under correlation shifts and enhances training performance in scenarios with correlated features. However, to our knowledge, there has been no previous work focusing on learning disentangled representations or exploring MI estimation for visual quality assessment.

## 3  Mutual Information Estimation and Minimization

Given the content-aware and distortion-aware representations $(\mathbf{x}, \mathbf{y})$, our goal is to estimate the MI between $\mathbf{x}$ and $\mathbf{y}$ and further minimize it. In this section, we explain the mathematical background of how to leverage a neural network to estimate the MI between $\mathbf{x}$ and $\mathbf{y}$.

The MI between $\mathbf{x}$ and $\mathbf{y}$ can be defined as:

$$
\begin{aligned}
\mathcal{I}(\mathbf{x}; \mathbf{y}) &= \int p(\mathbf{x}, \mathbf{y}) \log \frac{p(\mathbf{x}, \mathbf{y})}{p(\mathbf{x})p(\mathbf{y})} d\mathbf{x}d\mathbf{y} \\
&= \mathbb{E}_{p(\mathbf{x},\mathbf{y})} \left[ \log \frac{p(\mathbf{x}, \mathbf{y})}{p(\mathbf{x})p(\mathbf{y})} \right]
\end{aligned}
\tag{1}
$$

where $p(\mathbf{x}, \mathbf{y})$ is the joint distribution, $p(\mathbf{x})$ and $p(\mathbf{y})$ are the marginal distributions.

Unfortunately, the exact computation of MI between high-dimensional representations is actually intractable. Therefore, inspired by [6, 8, 15], we focus on estimating the MI upper bound and further minimize it. The tight upper bound of mutual information (MI) means an upper boundary that is always higher the actual value of MI. A tight upper bound means the bound is close to the actual value of MI and equal to MI under certain conditions. An MI upper bound estimator $\hat{\mathcal{I}}(\mathbf{x}; \mathbf{y})$ can be formulated as (proof in Appendix A):

$$
\hat{\mathcal{I}}(\mathbf{x}; \mathbf{y}) := \mathbb{E}_{p(\mathbf{x},\mathbf{y})}[\log p(\mathbf{y}|\mathbf{x})] - \mathbb{E}_{p(\mathbf{x})}\mathbb{E}_{p(\mathbf{y})}[\log p(\mathbf{y}|\mathbf{x})]
\tag{2}
$$

Since the conditional distribution $p(\mathbf{y}|\mathbf{x})$ is unavailable in our case, we approximate it using a variational distribution $q_\phi(\mathbf{y}|\mathbf{x}) = \mathcal{Q}_\phi(\mathbf{x}, \mathbf{y})$ , where the conditional distribution is inferred by another light neural network $\mathcal{Q}_\phi$ with parameters $\phi$. Then the variational form $\hat{\mathcal{I}}_\mathrm{v}(\mathbf{x}; \mathbf{y})$ can be formulated as (in a discretized form):

$$
\hat{\mathcal{I}}_\mathrm{v}(\mathbf{x}; \mathbf{y}) = \frac{1}{N^2} \sum_{i=1}^{N} \sum_{j=1}^{N} \left[ \log q_\phi\left(\mathbf{y}_i|\mathbf{x}_i\right) - \log q_\phi\left(\mathbf{y}_j|\mathbf{x}_i\right) \right]
\tag{3}
$$

where $\{(\mathbf{x}_i, \mathbf{y}_i)\}_{i=1}^{N}$ is $N$ samples pairs drawn from the joint distribution $p(\mathbf{x}, \mathbf{y})$. To make $\hat{\mathcal{I}}_\mathrm{v}(\mathbf{x}; \mathbf{y})$ a tight MI upper bound, $\mathcal{Q}_\phi$ is trained to accurately approximate $p(\mathbf{y}|\mathbf{x})$ by minimizing the KL divergence between $p(\mathbf{y}|\mathbf{x})$ and $q_\phi(\mathbf{y}|\mathbf{x})$:

$$
\begin{aligned}
&\min_\phi \mathrm{KL}\left(p(\mathbf{y}|\mathbf{x}) \| q_\phi(\mathbf{y}|\mathbf{x})\right) \\
&= \min_\phi \underbrace{\mathbb{E}_{p(\mathbf{x},\mathbf{y})}[\log p(\mathbf{y}|\mathbf{x})]}_{\text{No relation with } \phi} - \underbrace{\mathbb{E}_{p(\mathbf{x},\mathbf{y})}\left[\log q_\phi(\mathbf{y}|\mathbf{x})\right]}_{\text{to be minimized}}
\end{aligned}
\tag{4}
$$

Obviously, the first term in Equation 4 has no relation with $\phi$, thus Equation 4 equals minimization of the second term. Therefore, the can be a tight MI upper bound if we minimize the following negative log-likelihood of the inferred conditional distribution:

$$
\mathcal{L}_{\mathrm{MI}} = -\frac{1}{N} \sum_{i=1}^{N} \log q_\phi(\mathbf{y}_i|\mathbf{x}_i) = -\frac{1}{N} \sum_{i=1}^{N} \log \mathcal{Q}_\phi(\mathbf{x}_i, \mathbf{y}_i)
\tag{5}
$$

Now, given the representations $\mathbf{x}$ and $\mathbf{y}$, we can train the MI estimator $\hat{\mathcal{I}}_\mathrm{v}(\mathbf{x}; \mathbf{y})$ to predict the MI between $\mathbf{x}$ and $\mathbf{y}$ by minimizing $\mathcal{L}_{\mathrm{MI}}$. Afterwards, we minimize $\hat{\mathcal{I}}_\mathrm{v}(\mathbf{x}; \mathbf{y})$ for explicit disentanglement, detailed implementations will be explained in Section 4.4.

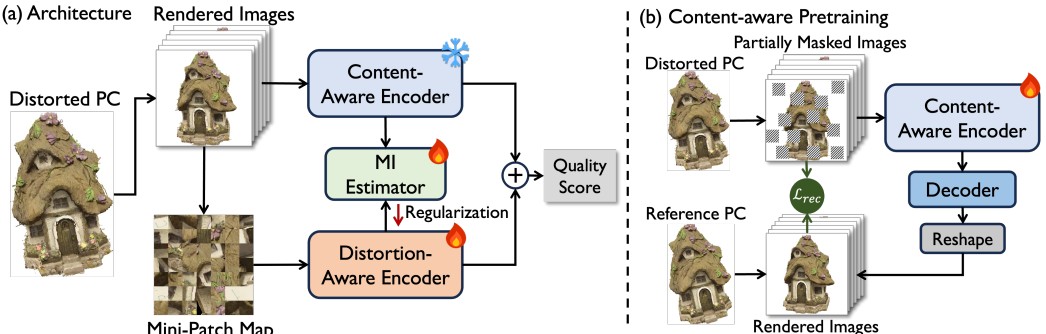

Figure 2: Architecture of proposed DisPA (a). Our DisPA consists of two encoders $\mathcal{F}$ and $\mathcal{G}$ for learning content-aware and distortion-aware representations, and an MI estimator $\mathcal{M}$. The content-aware encoder $\mathcal{F}$ is pretrained using masked autoencoding (b). "$\bigoplus$" denotes concatenation.

## 4 Proposed Framework

### 4.1 Overall Architecture

The aforementioned analysis in Section 1 reveals that HVS processes high-level and low-level features in a relatively separate manner. To obey this mechanism, as illustrated in Figure 2 (a), the architecture of DisPA is divided into two branches to learn content-aware and distortion-aware representations, respectively. Given a distorted point cloud $P$, we first render it into multi-view images $I$. The multi-view images are fed into a frozen pretrained vision transformer (ViT) [10] $\mathcal{F}$ with parameters $\Theta_f$ to generate the content-aware representation $\mathbf{x}$. Next, the multi-view images are decomposed into a mini-patch map $M$ through grid mini-patch sampling. The mini-patch map is encoded by the distortion-aware encoder $\mathcal{G}$ (a Swin Transformer) [24]) with parameters $\Theta_g$ to obtain representation $\mathbf{y}$. After obtaining $\mathbf{x}$ and $\mathbf{y}$, we also use them to train the MI estimator $\mathcal{M}$ and obtain the estimated MI $\hat{\mathcal{I}}_v(\mathbf{x}; \mathbf{y})$ following the process in Section 3. Finally, we concatenate $\mathbf{x}$ and $\mathbf{y}$ (denoted as $[\cdot, \cdot]$) and regress it by fully-connected layers $\mathcal{H}$ with parameters $\Theta_h$ to predict quality score $\hat{q}$. The whole process can be described as follows:

$$\hat{q} = \mathcal{H}([\mathcal{F}(I; \Theta_f), \mathcal{G}(M; \Theta_g)]; \Theta_h) \tag{6a}$$

$$\hat{\mathcal{I}}_v(\mathbf{x}; \mathbf{y}) = \mathcal{M}(\mathcal{F}(I; \Theta_f), \mathcal{G}(M; \Theta_g)) \tag{6b}$$

### 4.2 Content-Aware Pretraining via Masked Autoencoding

As analyzed in Section 1, the learning difficulty of content representation is more intractable than distortion due to the limited dataset scale in terms of point cloud content. To address this problem, we pretrain the content-aware encoder $\mathcal{F}$ via the proposed masked autoencoding strategy. As illustrated in Figure 2 (b), given a distorted point cloud $P$ and its corresponding reference point cloud $P_{\text{ref}}$, our goal is to render $P$ and $P_{\text{ref}}$ into multi-view images $\{I^{(n)} \in \mathbb{R}^{H \times W \times 3}\}_{n=1}^{N_v}$, $\{I_{\text{ref}}^{(n)} \in \mathbb{R}^{H \times W \times 3}\}_{n=1}^{N_v}$ and pretrain $\mathcal{F}$ by using partially masked $I^{(n)}$ to reconstruct $I_{\text{ref}}^{(n)}$, where $N_v$ is the number of views.

**Multi-View Rendering.** Instead of directly performing masked autoencoding in 3D space, we render point clouds into 2D images to achieve pixel-to-pixel correspondence between the rendered images of $P$ and $P_{\text{ref}}$, which facilitates the computation of pixel-wise reconstruction loss between the predicted patches and the ground truth patches. To perform the rendering, we translate $P$ (or $P_{\text{ref}}$) to the origin and geometrically normalize it to the unit sphere to achieve a consistent spatial scale. Then, to fully capture the quality information of 3D point clouds, we apply random rotations before rendering $P$, $P_{\text{ref}}$ into $\{I^{(n)}\}_{n=1}^{N_v}$ and $\{I_{\text{ref}}^{(n)}\}_{n=1}^{N_v}$.

**Patchifying and Masking.** After obtaining the rendered image $I^{(n)}$, we partition it into non-overlapping $16 \times 16$ patches following [10]. Then we randomly sample a subset of patches and mask the remaining ones, where the masking ratio is relaxed to 50% instead of the high ratio in [12] (*e.g.,*

75% and even higher) because some point cloud samples in PCQA datasets exhibit severe distortions, necessitating more patches to extract effective content-aware information. In addition, the relatively low masking ratio can mitigate the influence of the background of $I^{(n)}$.

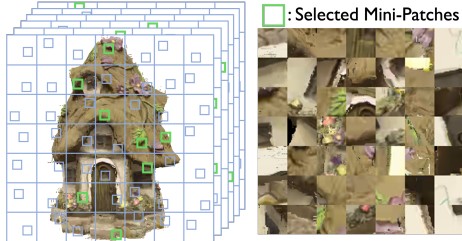

Figure 3: Illustration of mini-patch map generation.

**Encoding and Reconstruction.** The unmasked patches of $I^{(n)}$ are fed into the ViT $\mathcal{F}$ for initial embedding and subsequent encoding to obtain the representation $\mathbf{x}$. To compensate for the scarcity of point cloud content in PCQA datasets, we initialize the encoder using the parameters optimized on ImageNet-1K [9]. Next, to reconstruct $I_{\text{ref}}^{(n)}$, we feed $\mathbf{x}$ into a decoder and reshape it into 2D pixels to generate the reconstructed $\hat{I}_{\text{ref}}^{(n)}$. By reconstructing masked reference patches from unmasked distorted patches, the encoder $\mathcal{F}$ is forced to focus more on semantic information than distortion patterns. The content-aware representation can be learned by the reconstruction loss $\mathcal{L}_{\text{rec}}$:

$$\mathcal{L}_{\text{rec}} = \sum_{n=1}^{N_v} \left\| \hat{I}_{\text{ref}}^{(n)} - I_{\text{ref}}^{(n)} \right\|_2^2 \tag{7}$$

### 4.3 Distortion-Aware Mini-patch Map Generation

To learn an effective distortion-aware representation $\mathbf{y}$, we decompose the multi-view images $\{I^{(n)}\}_{n=1}^{N_v}$ to a mini-patch map through grid mini-patch sampling, following [40, 42, 52]. As illustrated in Figure 3, the distortion pattern is well preserved and even more obvious on the mini-patch map while the content information is blurred. More concretely, for each multi-view image $I^{(n)}$, we first split it into uniform $L \times L$ grids, the set of grids $G^{(n)}$ can be described as:

$$G^{(n)} = \{g_{0,0}^{(n)}, \cdots, g_{i,j}^{(n)}, \cdots, g_{L,L}^{(n)}\}, \quad g_{i,j}^{(n)} = I^{(n)}[\frac{i \times H}{L} : \frac{(i+1) \times H}{L}, \frac{j \times W}{L} : \frac{(j+1) \times W}{L}] \tag{8}$$

where $g_{i,j}^{(n)} \in \mathbb{R}^{\frac{H}{L} \times \frac{W}{L} \times 3}$ denotes the grid in the $i$-th row and $j$-th column of $I^{(n)}$. Then we sample the mini-patches from each $g_{i,j}^{(n)}$ and splice all the selected mini-patches to get the mini-patch map $M$. Note that blank mini-patches (*i.e.*, image background) are ignored, and the map is ensured to $M \in \mathbb{R}^{H \times W \times 3}$ by filling in the unemployed mini-patches. After the mini-patch map generation, we feed it into the distortion-aware encoder $\mathcal{G}$ to generate the corresponding representation $\mathbf{y}$.

### 4.4 Disentangled Representation Learning

**MI-based Regularization.** After obtaining content-aware and distortion-aware representations $\mathbf{x}$ and $\mathbf{y}$, we further disentangle them by minimizing the MI upper bound in $\hat{\mathcal{I}}_v(\mathbf{x}; \mathbf{y})$ in Equation 3. As revealed in Equation 4 and 5, the key to accurately estimate a tight $\hat{\mathcal{I}}_v(\mathbf{x}; \mathbf{y})$ is to minimize the negative log-likelihood of the variational network $Q_\phi(\mathbf{x}, \mathbf{y})$. Here we implement the $\mathcal{Q}_\phi$ using MLPs, and model the variational distribution as an isotropic Gaussian parameterized by a mean value $\boldsymbol{\mu}_\phi = [\mu_\phi(x_1), ..., \mu_\phi(x_D)]$ and a diagonal covariance matrix $\boldsymbol{\Sigma} = \boldsymbol{\sigma}_\phi^2 \boldsymbol{I}$, where $\boldsymbol{\sigma}_\phi = [\sigma_\phi(x_1), ..., \sigma_\phi(x_D)]$, $D$ is the feature dimension of $\mathbf{x}$ and $\mathbf{y}$. Then the variational distribution can be inferred as:

$$q_\phi(\mathbf{y}|\mathbf{x}) = \mathcal{Q}_\phi(\mathbf{x}, \mathbf{y}) = \prod_{d=1}^{D} \frac{1}{\sqrt{(2\pi)^D \sigma_\phi^2(x_d)}} \exp\left\{ -\frac{(y_d - \mu_\phi(x_d))^2}{2\sigma_\phi^2(x_d)} \right\} \tag{9}$$

where $\boldsymbol{\mu}_\phi$ and $\boldsymbol{\sigma}_\phi^2$ are obtained via the last two MLP layers. $\phi$ is optimized by minimizing $\mathcal{L}_{\text{MI}}$ in Equation 5, the negative log-likelihood of $\mathcal{Q}_\phi(\mathbf{x}, \mathbf{y})$. It is noted that the parameters of $\phi$ are optimized independently with the main networks $\Theta_f$ and $\Theta_g$, seeing Algorithm 1.

**Loss Function.** After obtaining the $\hat{\mathcal{I}}_v(\mathbf{x}; \mathbf{y})$, we incorporate it into our total training objective as a regularizer to disentangle the content-aware and distortion-aware representations, the total training

**Algorithm 1** Disentangled Representation Learning Pipeline

---

**Input:** A batch of rendered images $\{I_b^{(n)}|_{n=1}^{N_v}\}_{b=1}^B$; mini-patch map $\{M_b\}_{b=1}^B$; networks $\mathcal{F}, \mathcal{G}, \mathcal{H}$
  with parameters $\Theta_f', \Theta_g, \Theta_h$; MI estimator $\mathcal{M}$ with variational network $\mathcal{Q}_\phi$; optimizer; $\lambda_1, \lambda_2$
**Output:** Updated parameters $\Theta_g', \Theta_h'$ and $\phi'$                  *// Parameters $\Theta_f'$ is frozen*

1: Encode rendered images to generate content-aware representation $\mathbf{x} \leftarrow \mathcal{F}(\{I_b^{(n)}\}); \Theta_f')$
2: Encode mini-patch map to generate distortion-aware representation $\mathbf{y} \leftarrow \mathcal{G}(\{M_b\}; \Theta_g)$
3: **for** $m = 1 \rightarrow N_\mathcal{M}$ **do**                *// Update the MI estimator by training $\mathcal{Q}_\phi$*
4:      Compute negative log-likelihood $\mathcal{L}_{\text{MI}} \leftarrow \sum_{b=1}^B \mathcal{Q}_\phi(\mathbf{x}, \mathbf{y})$
5:      Update $\phi$ by minimizing $\mathcal{L}_{\text{MI}}$    $\phi' \leftarrow$ optimizer $(\phi, \nabla_\phi \mathcal{L}_{\text{MI}})$
6: Compute the estimated MI $\hat{I}_v(\mathbf{x}; \mathbf{y}) \leftarrow \frac{1}{B^2} \sum_{i=1}^B \sum_{j=1}^B [\log \mathcal{Q}_{\phi'}(\mathbf{x}_i, \mathbf{y}_i) - \log \mathcal{Q}_{\phi'}(\mathbf{x}_i, \mathbf{y}_j)]$
7: Predict the quality scores $\hat{q}_b \leftarrow \mathcal{H}([\mathbf{x}_b, \mathbf{y}_b]; \Theta_h)$
8: Compute the total loss $\mathcal{L} \leftarrow \frac{1}{B} \sum_{b=1}^B (\hat{q}_b - q_b)^2 + \lambda_1 \mathcal{L}_{\text{rank}} + \lambda_2 \hat{I}_v(\mathbf{x}; \mathbf{y})$
9: Update the parameters $\{\Theta_g', \Theta_h'\} \leftarrow$ optimizer$(\{\Theta_g, \Theta_h\}, \{\nabla_{\Theta_g}\mathcal{L}, \nabla_{\Theta_h}\mathcal{L}\})$

---

loss function $\mathcal{L}$ can be formulated as:

$$\mathcal{L} = \frac{1}{B} \sum_{b=1}^B (\hat{q}_b - q_b)^2 + \lambda_1 \mathcal{L}_{\text{rank}} + \lambda_2 \hat{I}_v(\mathbf{x}; \mathbf{y}) \tag{10}$$

where $B$ is the batch size and $\mathcal{L}_{\text{rank}}$ is a differential ranking loss following [51, 33]. The $\lambda_1$ and $\lambda_2$ are weighting factors to balance each loss term. To better recognize quality differences for the point clouds with close MOSs, the differential ranking loss [51] $\mathcal{L}_{\text{rank}}$ is used to model the ranking relationship between $\hat{q}$ and $q$:

$$\mathcal{L}_{rank} = \frac{1}{B^2} \sum_{i=1}^B \sum_{j=1}^B \max(0, |q_i - q_j| - e(q_i, q_j) \cdot (\hat{q}_i - \hat{q}_j)),$$

$$e(q_i, q_j) = \left\{ \begin{array}{c} 1, q_i \geq q_j \\ -1, q_i < q_j \end{array} \right. \tag{11}$$

Algorithm 1 summarizes the overall pipeline of the disentangled representation learning framework (one iteration), where $N_\mathcal{M}$ is the steps of updating for variational networks per epoch, and $\mathcal{F}$ is initialized with the pretrained parameters $\Theta_f$ after masked autoencoding. The parameters of the main network $\Theta_g, \Theta_h$ and the variational networks $\phi$ are updated alternately.

## 5 Experiments

### 5.1 Datasets and Evaluation Metrics

**Datasets.** Our experiments are performed on three popular PCQA datasets, including LS-PCQA [23], SJTU-PCQA [46], and WPC [20]. The content-aware pretraining is based on LS-PCQA, which contains 24,024 distorted point clouds, and each reference point cloud is impaired with 33 types of distortions (*e.g.*, V-PCC, G-PCC) under 7 levels. The disentangled representation learning is conducted on all three datasets separately using labeled data, where SJTU-PCQA includes 9 reference point clouds and 378 distorted samples impaired with 7 types of distortions (*e.g.*, color noise, downsampling) under 6 levels, while WPC contains 20 reference point clouds and 740 distorted samples disturbed by 5 types of distortions (*e.g.*, compression, gaussian noise).

**Evaluation Metrics.** Three widely adopted evaluation metrics are employed to quantify the level of agreement between predicted quality scores and ground truth (*i.e.*, Mean Opinion Score, MOS): Spearman rank order correlation coefficient (SROCC), Pearson linear correlation coefficient (PLCC), and root mean square error (RMSE). To ensure consistency between the value ranges of the predicted scores and subjective values, nonlinear Logistic-4 regression is used to align their ranges.

Table 1: Quantitative comparison of the state-of-the-art methods and proposed DisPA on LS-PCQA [23], SJTU-PCQA [46], WPC [20]. The best results are shown in **bold**, and second results are underlined. "P" / "I" stands for the method is based on the point cloud/image modality, respectively. "↑" / "↓" indicates that larger/smaller is better.

| Ref | Modal | Method | LS-PCQA | | | SJTU-PCQA | | | WPC | | |
|---|---|---|---|---|---|---|---|---|---|---|---|
| | | | SROCC ↑ | PLCC ↑ | RMSE ↓ | SROCC ↑ | PLCC ↑ | RMSE ↓ | SROCC ↑ | PLCC ↑ | RMSE ↓ |
| FR | P | MSE-p2po [26] | 0.325 | 0.528 | 0.158 | 0.783 | 0.845 | 0.122 | 0.564 | 0.557 | 0.188 |
| | P | HD-p2po [26] | 0.291 | 0.488 | 0.163 | 0.681 | 0.748 | 0.156 | 0.106 | 0.166 | 0.222 |
| | P | MSE-p2pl [36] | 0.311 | 0.498 | 0.160 | 0.703 | 0.779 | 0.149 | 0.445 | 0.491 | 0.199 |
| | P | HD-p2pl [36] | 0.291 | 0.478 | 0.163 | 0.617 | 0.661 | 0.177 | 0.344 | 0.380 | 0.211 |
| | P | PSNR-yuv [36] | 0.548 | 0.547 | 0.155 | 0.704 | 0.715 | 0.165 | 0.563 | 0.579 | 0.186 |
| | P | PointSSIM [1] | 0.180 | 0.178 | 0.183 | 0.735 | 0.747 | 0.157 | 0.453 | 0.481 | 0.200 |
| | P | PCQM [27] | 0.439 | 0.510 | **0.152** | 0.864 | 0.883 | 0.112 | 0.750 | 0.754 | 0.150 |
| | P | GraphSIM [48] | 0.320 | 0.281 | 0.178 | 0.856 | 0.874 | 0.114 | 0.679 | 0.693 | 0.165 |
| | P | MS-GraphSIM [49] | 0.389 | 0.348 | 0.174 | 0.888 | 0.914 | 0.096 | 0.704 | 0.718 | 0.159 |
| NR | I | PQA-Net [21] | 0.588 | 0.592 | 0.202 | 0.659 | 0.687 | 0.172 | 0.547 | 0.579 | 0.189 |
| | I | IT-PCQA [47] | 0.326 | 0.347 | 0.224 | 0.539 | 0.629 | 0.218 | 0.422 | 0.468 | 0.221 |
| | P | GPA-Net [32] | 0.592 | 0.619 | 0.186 | 0.878 | 0.886 | 0.122 | 0.758 | 0.769 | 0.162 |
| | P | ResSCNN [23] | 0.594 | 0.624 | 0.172 | 0.834 | 0.863 | 0.153 | 0.735 | 0.752 | 0.177 |
| | P+I | MM-PCQA [51] | 0.581 | 0.597 | 0.189 | 0.876 | 0.898 | 0.109 | 0.761 | 0.774 | 0.149 |
| | I | CoPA [33] | 0.621 | **0.636** | 0.161 | 0.897 | 0.913 | 0.092 | 0.779 | 0.785 | 0.144 |
| | I | **DisPA (ours)** | **0.625** | 0.631 | 0.160 | **0.908** | **0.919** | **0.089** | **0.788** | **0.790** | **0.138** |

## 5.2 Implementation Details

Our experiments are performed using PyTorch [28] on NVIDIA 3090 GPUs. All point clouds are rendered into 6-view projected images with a spatial resolution of $512 \times 512$ by PyTorch3D [30]. The encoders $\mathcal{F}$ and $\mathcal{G}$ are ViT-B [10] and Swin-T [24], respectively.

**Content-Aware Pretraining.** The pretraining is performed for 200 epochs, the initial learning rate is 3e-4, and the batch size is 64 by default. Adam optimizer [18] is employed with weight decay of 0.0001. Each point cloud is randomly rotated 6 times before being rendered into 6-view images to fully take advantage of quality information of point clouds.

**Disentangle Representation Learning.** The steps of updating $N_{\mathcal{M}}$ of MI estimator is set to 10. We use the Adam optimizer with weight decay of 0.0001 and batch size of 16. The hidden dimension of fully-connected layers is set to 64. The learning rate is initialized with 0.003 and decayed by 0.95 exponentially per epoch. For LS-PCQA, the model is trained for 20 epochs, while 150 epochs for SJTU-PCQA and WPC. The hyper-parameter $\lambda_1, \lambda_2$ is set to 1 and 0.01 according to the loss scales.

**Data Split.** Considering the limited dataset scale, in the training stage, 5-fold cross validation is used for SJTU-PCQA and WPC to reduce content bias. Take SJTU-PCQA for example, in each fold, the dataset is split into train-test with ratio 7:2 according to the reference point clouds, where the performance on testing set with minimal training loss is recorded and then averaged across five folds to get the final result. A similar procedure is repeated for WPC where the train-test ratio is 4:1. As for the large-scale LS-PCQA, it is split into train-val-test with a ratio around 8:1:1 (no content overlap exists). The result on the testing set with the best validation performance is recorded. Note that the pretraining is only conducted on the training set of LS-PCQA.

## 5.3 Comparison with State-of-the-art Methods

15 state-of-the-art PCQA methods are selected for comparison, including 9 FR-PCQA and 6 NR-PCQA methods. For a comprehensive comparison, we conduct the experiment in four aspects: 1) We compare a quantitative comparison of prediction accuracy, following the cross-validation configuration in Section 5.2. 2) We perform the statistical analysis in Figure 1 for DisPA. 3) We present qualitative examples to demonstrate the superiority of our model in terms of avoiding overfitting when point content varies. 4) We report the results of cross-dataset evaluation for the NR-PCQA methods to verify the generalizability of our model.

**Quantitative Comparison.** The prediction accuracy of all selected methods are presented in Table 1, which demonstrates the competitive performance of proposed DisPA across all three datasets, and outperforms all the FR-PCQA methods on SJTU-PCQA and WPC. Compared with the NR-PCQA methods, DisPA outperforms CoPA [33] by about $1.3\%$ in terms of SROCC on LS-PCQA, and

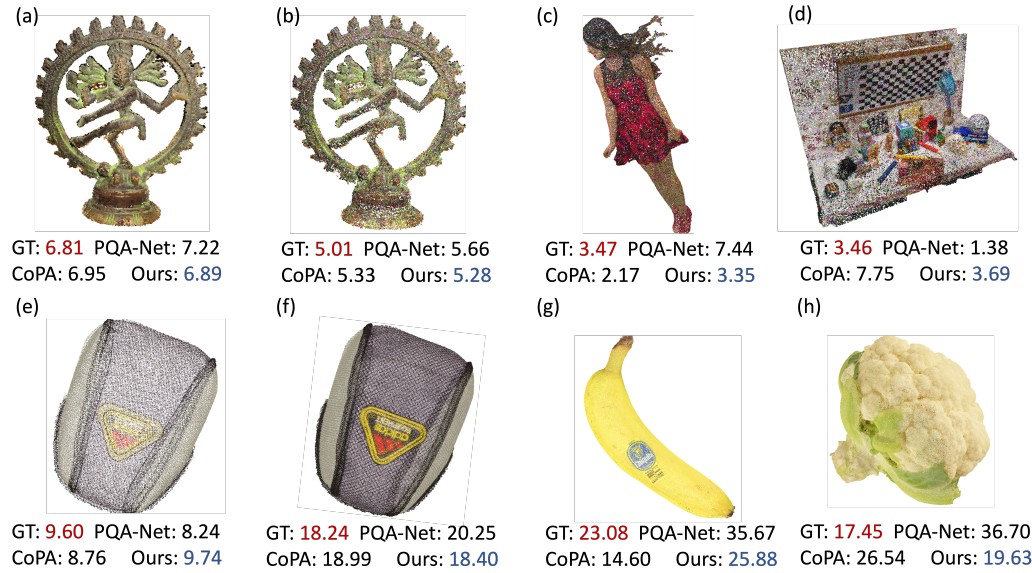

| | | | |
|---|---|---|---|
| (a) GT: 6.81 PQA-Net: 7.22 CoPA: 6.95 Ours: 6.89 | (b) GT: 5.01 PQA-Net: 5.66 CoPA: 5.33 Ours: 5.28 | (c) GT: 3.47 PQA-Net: 7.44 CoPA: 2.17 Ours: 3.35 | (d) GT: 3.46 PQA-Net: 1.38 CoPA: 7.75 Ours: 3.69 |
| (e) GT: 9.60 PQA-Net: 8.24 CoPA: 8.76 Ours: 9.74 | (f) GT: 18.24 PQA-Net: 20.25 CoPA: 18.99 Ours: 18.40 | (g) GT: 23.08 PQA-Net: 35.67 CoPA: 14.60 Ours: 25.88 | (h) GT: 17.45 PQA-Net: 36.70 CoPA: 26.54 Ours: 19.63 |

Figure 5: Qualitative Evaluation of NR-PCQA methods (PQA-Net [21], CoPA [33] and DisPA) on SJTU-PCQA [46] and WPC [20]. Figure (b)-(d) share the same distortion pattern (*i.e.*, color noise), same for (f)-(h) (*i.e.*, downsampling). "GT" denotes ground truth.

1.3% on SJTU-PCQA. Note that CoPA has also been pretrained on LS-PCQA. Furthermore, DisPA significantly reduces RMSE by 4.2% compared to CoPA.

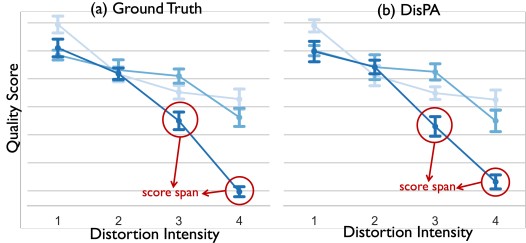

Figure 4: Statistical Analysis of SJTU-PCQA (part) and predicted quality scores of DisPA.

**Statistical Analysis.** We perform the statistical analysis on SJTU-PCQA in Figure 4 for DisPA. Compared with the statistics of PQA-Net and GPA-Net in Figure 1, our DisPA not only predicts quality scores more accurately, but also obviously predicts closer score spans when point cloud content varies, even when the distortion intensity is at the highest level. The statistical analysis demonstrates that the content-aware pretraining strategy can effectively address the problem of superior difficulty of learning representations for point cloud content caused by data imbalance.

**Qualitative Evaluation.** In Figure 5, we present examples of SJTU-PCQA and WPC with predicted scores of PQA-Net [21] and CoPA [33], where Figure 5 (a)(b), (e)(f) share the same content, (b)-(d), (f)-(h) share the the same distortion. Note that each score is predicted on the testing set of 5-fold validation. We can see that the predicted score of our DisPA is obviously closer to the ground truth (*i.e.*, Mean Opinion Score, MOS) and dose not deviate from the MOS when content varies, which further validates the effectiveness of content-aware pretraining and representation disentanglement.

**Cross-Dataset Validation.** To test the generalization capability of NR-PCQA methods when encountering various data distribution, we perform cross-dataset on LS-PCQA [23], SJTU-PCQA [46] and WPC [20]. In Table 2, we mainly train the compared models on the complete LS-PCQA and test the trained model on the complete SJTU-PCQA and WPC, and the result with minimal training loss is recorded. This procedure is repeated for mutual cross-dataset validation between SJTU-PCQA and WPC. From Table 2, we can see that the performance of the cross-dataset validation is relatively low due to the tremendous variation of data distribution. However, our method still present competitive performances, demonstrating the superior generalizability of DisPA.

Table 2: Cross-dataset validation on LS-PCQA [23], SJTU-PCQA [46] and WPC [20] (complete set). The best results (PLCC) are in **bold**, and the second results are underlined.

| Train | Test | PQA-Net | GPA-Net | MM-PCQA | CoPA | **DisPA** |
|---|---|---|---|---|---|---|
| LS | SJTU | 0.342 | 0.556 | 0.581 | 0.644 | **0.653** |
| LS | WPC | 0.266 | 0.433 | 0.454 | **0.516** | 0.505 |
| WPC | SJTU | 0.235 | 0.553 | 0.612 | 0.643 | **0.657** |
| SJTU | WPC | 0.220 | 0.418 | 0.269 | 0.533 | **0.535** |

Table 3: Ablation study of DisPA on SJTU-PCQA [46]. The best results are in **bold**.

| Index | Pretraining | Disentanglement | Loss | SROCC | PLCC |
|---|---|---|---|---|---|
| ① | ✓ | ✓ | ✓ | **0.908** | **0.919** |
| ② | ✗ | ✓ | ✓ | 0.876 | 0.897 |
| ③ | ✓ | ✗ | ✓ | 0.849 | 0.865 |
| ④ | ✓ | Cos. Similarity | ✓ | 0.884 | 0.910 |
| ⑤ | ✓ | Con. only | ✓ | 0.811 | 0.832 |
| ⑥ | ✓ | Dis. only | ✓ | 0.763 | 0.778 |
| ⑦ | ✓ | ✓ | $w/o\ \mathcal{L}_{rank}$ | 0.892 | 0.910 |

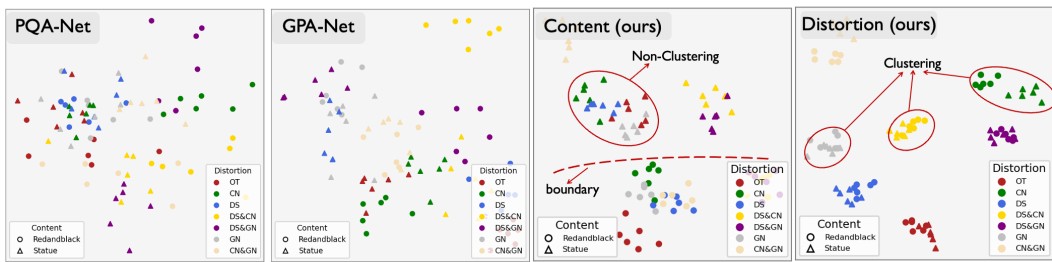

Figure 6: Visualization of t-SNE embeddings of representations of PQA-Net, GPA-Net, content-aware and distortion-aware branches of our DisPA.

## 5.4 Ablation Study

We conduct ablation study of DisPA on SJTU-PCQA [46] in Table 3. From Table 3, we have following observations: 1) Seeing ① and ②, the pretraining strategy effectively improves the performance of DisPA. 2) Seeing ①, ③ and ④, the philosophy of representation disentanglement brings significant improvements to our model, because using simple cosine similarity in ④ for disentanglement can achieve fair performance. However, using MI for disentanglement can better constrain the dependence between representations. 3) Seeing ①, ⑤ and ⑥, using single branch to infer quality scores causes poor performance, since PCQA is a combination judgement based on the interaction of distortion estimation and content recognition. 4) Seeing ① and ⑦, the performance is close, demonstrating the robustness of our model using different training loss functions.

Furthermore, as shown in Figure 6, we conduct a t-SNE visualization to compare the representation embeddings of PQA-Net, GPA-Net, content-aware and distortion-aware branches of our DisPA on the testing set of SJTU-PCQA. PQA-Net and GPA-Net are selected for comparison because these two methods both use distortion type prediction to learn distortion-aware representations. The scattered points are color and shape marked according to distortion type and content. The distortion-aware features are visualized in 3rd sub-image, where we can see that the learned distortion-aware representation shows clear and separate clustering for different distortion types, indicating a strong correlation with degradations. The content-aware features present non-clustering for distortion types but a clear boundary to split the content.

## 6 Conclusion

In this paper, we propose a disentangled representation learning framework (DisPA) for No-Reference Point Cloud Quality Assessment (NR-PCQA) based on mutual information (MI) minimization. As for the MI minimization, we use a variational network to infer the upper bound of the MI and further minimize it to achieve explicit representation disentanglement. In addition, to tackle the nontrivial learning difficulty of content-aware representations, we propose a novel content-aware pretraining strategy to enable the encoder to capture effective semantic information from distorted point clouds. Furthermore, to learn effective distortion-aware representations, we decompose the rendered images into mini-patch maps, which can preserve original distortion pattern and make the encoder ignore the global content. We demonstrate the high performance of DisPA on three popular PCQA benchmarks and validate the generalizability compared with multiple NR-PCQA models.

**Acknowledgments.** This paper is supported in part by National Natural Science Foundation of China (62371290, U20A20185), the Fundamental Research Funds for the Central Universities of China, and 111 project (BP0719010).

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

# A  Proof of Theorems

*Proof of Equation 2.* To show that $\hat{\mathcal{I}}(\mathbf{x}; \mathbf{y})$ is an upper bound of $\mathcal{I}(\mathbf{x}; \mathbf{y})$, we calculate their difference:

$$\hat{\mathcal{I}}(\mathbf{x}; \mathbf{y}) - \mathcal{I}(\mathbf{x}; \mathbf{y}) = \underbrace{\mathbb{E}_{p(\mathbf{x},\mathbf{y})}[\log p(\mathbf{y}|\mathbf{x})] - \mathbb{E}_{p(\mathbf{x})}\mathbb{E}_{p(\mathbf{y})}[\log p(\mathbf{y}|\mathbf{x})]}_{\text{Definition of } \hat{\mathcal{I}}(\mathbf{x}; \mathbf{y}) \text{ in Equation 2}}$$

$$- \underbrace{\mathbb{E}_{p(\mathbf{x},\mathbf{y})}[\log p(\mathbf{y}|\mathbf{x}) - \log p(\mathbf{y})]}_{\text{Definition of } \mathcal{I}(\mathbf{x}; \mathbf{y}) \text{ in Equation 1}}$$

$$= \mathbb{E}_{p(\mathbf{x},\mathbf{y})}[\log p(\mathbf{y})] - \mathbb{E}_{p(\mathbf{x})}\mathbb{E}_{p(\mathbf{y})}[\log p(\mathbf{y}|\mathbf{x})] \tag{12}$$

Since $\log p(\mathbf{y})$ has no relation with $\mathbf{x}$, so $\mathbb{E}_{p(\mathbf{x},\mathbf{y})}[\log p(\mathbf{y})] = \mathbb{E}_{p(\mathbf{y})}[\log p(\mathbf{y})]$. Then the equation can be formulated as:

$$\hat{\mathcal{I}}(\mathbf{x}; \mathbf{y}) - \mathcal{I}(\mathbf{x}; \mathbf{y}) = \mathbb{E}_{p(\mathbf{y})}\left[\log p(\mathbf{y}) - \mathbb{E}_{p(\mathbf{x})}[\log p(\mathbf{y}|\mathbf{x})]\right]. \tag{13}$$

Recalling that the marginal distribution can obtained by integrating the joint distribution over the values of the other random variables:

$$p(\mathbf{y}) = \int p(\mathbf{x}, \mathbf{y})\mathrm{d}\mathbf{x} = \int p(\mathbf{y}|\mathbf{x})p(\mathbf{x})\mathrm{d}\mathbf{x} = \mathbb{E}_{p(\mathbf{x})}[p(\mathbf{y}|\mathbf{x})] \tag{14}$$

Note that $\log(\cdot)$ is a concave function, by Jensen's Inequality, we have

$$\underbrace{\log p(\mathbf{y}) = \log\left(\mathbb{E}_{p(\mathbf{x})}[p(\mathbf{y}|\mathbf{x})]\right)}_{\text{Definition of marginal distribution in Equation 14}} \geq \mathbb{E}_{p(\mathbf{x})}[\log p(\mathbf{y}|\mathbf{x})] \tag{15}$$

Applying this inequality to Equation 13, we can conclude that $\hat{\mathcal{I}}(\mathbf{x}; \mathbf{y})$ is always greater than $\mathcal{I}(\mathbf{x}; \mathbf{y})$. Therefore, $\hat{\mathcal{I}}(\mathbf{x}; \mathbf{y})$ is an upper bound of $\mathcal{I}(\mathbf{x}; \mathbf{y})$. $\hat{\mathcal{I}}(\mathbf{x}; \mathbf{y}) = \mathcal{I}(\mathbf{x}; \mathbf{y})$ occurs only when $p(\mathbf{y}|\mathbf{x})$ holds the same for any $\mathbf{x}$, which means $\mathbf{x}$ and $\mathbf{y}$ are two totally independent representations.

# B  Limitations and Future Work

We designed a no-reference point cloud quality assessment (NR-PCQA) framework, whose experimental performances have been validated. However, our current design has two main limitations:

- Limitation of pretraining datasets. Ideally, the content-aware encoder $\mathcal{F}$ can be pretrained on a much larger dataset, despite the already large scale of LS-PCQA [23]. In our future work, we may pretrain it on distorted natural images or create a larger dataset of point clouds with various distortions.

- Estimation of mutual information (MI). Although using the current MI minimization strategy can achieve satisfactory results, we may make efforts in the future to work on a more efficient MI estimation method for high-dimensional representations.

