# OpenReview forum: "Learning Disentangled Representations for Perceptual Point Cloud Quality Assessment via Mutual Information Minimization"
_NeurIPS.cc/2024/Conference — NeurIPS 2024 poster_

### Official Review · Reviewer_nBew · 2024-07-01

**Soundness:** 2
**Presentation:** 2
**Contribution:** 2
**Rating:** 4
**Confidence:** 4

**Summary:**

This paper explores the method for No-Reference Point Cloud Quality Assessment. The key idea is to involve the disentangled representation learning to minimize mutual information between representations of point cloud content and distortion. The authors conduct experimental performance comparisons on three public databases and compare their proposed method with 15 existing models. The proposed method achieves optimal or suboptimal results in most of the metrics. The ablation study demonstrated the necessity
of each design part.

**Strengths:**

The paper is clearly written and provides a detailed formulation of the approach.

**Weaknesses:**

The weak point of the paper is its presentation. Many terms are introduced without explaining them properly, e.g. "the tight upper bound" or "the masked autoencoding strategy”. Furthermore, the Fig 2 is not well-designed with content aware branch. The Fig2 splits the proposed architecture and the content-aware pretraining and masked autoencoding strategy, which does not help the reader to understand architecture with figure. Lastly, it would be highly beneficial to make the code publicly available to enhance collaborative efforts and facilitate the sharing of this work. Apart from this, there are quite a few typos and grammar mistakes that should be corrected, such as “the can be”, “masked auto-encoding / autoencoding

**Questions:**

See Weaknesses.

**Limitations:**

See Weaknesses.

---

> ### Author Rebuttal · Authors · 2024-08-05
>
> **`R4-Weakness 1`**:  Thank you for your precious comments about presentation. We will provide more detailed explanations and make the paper easier to understand. The explanations of some terms are as follow:
>
> * The *tight upper bound* of mutual information (MI) means an upper boundary that is always higher the actual value of MI. A tight upper bound means the bound is close to the actual value of MI and equal to MI under certain conditions.
>
> * The *masked autoencoding (MAE)* strategy is a popular visual self-supervised learning scheme, which maskes some patches of the input image and training a model to reconstruct the masked patches. This method helps in learning robust representations of unlabeled images.
> * The *grid mini-patch sampling* means dividing an image into smaller, non-overlapping patches (mini-patches) arranged in a grid-like structure.
> * The *variational distribution* is a key concept in variational inference, a technique used in Bayesian statistics for approximating complex probability distributions. It is essentially a tractable distribution used to approximate the true posterior distribution of the model's latent variables
>
> **`R4-Weakness 2`**: Thank you for your constructive comments about the Fig 2, we have modified the Fig 2 following your suggestion. We split the architecture and the content-aware pretraining (i.e., the proposed masked autoencoding strategy) into two figures, and make some detailed modifications , as shown in `attached file Figure B` and `attached file Figure C`. This can make readers better understand the overall architecture and the pretraining strategy.
>
> **`R4-Weakness 3`**: Thank you for your comments about the code. The code is attached in the supplementary material (in the .zip file), and we will upload the code to Github to make the code publicly available.
>
> **`R4-Weakness 4`**: Thank you for comments about typos and grammar mistakes. We will carefully correct these detailed mistakes and ensure there will be no such mistakes in the final version.

---

> ### Comment · Area_Chair_F1UL · 2024-08-12
> **Would you please have a look at the author rebuttal?**
>
> Dear Reviewer,
>
> Thanks a lot for contributing to NeurIPS2024.
>
> The authors have provided detailed responses to your review. Would you please have a look at them at your earliest convenience?
>
> Thanks again. AC

---

### Official Review · Reviewer_hUdG · 2024-07-04

**Soundness:** 3
**Presentation:** 3
**Contribution:** 3
**Rating:** 6
**Confidence:** 4

**Summary:**

This paper proposes a novel no-reference quality assessment model tailored for point-cloud data. A disentangled representation learning strategy is leveraged to account for both content-aware information and distortion-aware information. Comprehensive experiments are conducted and the effectiveness of this proposed model is well verified.

**Strengths:**

The proposed framework is novel and the paper is easy to understand.

**Weaknesses:**

1. The idea of framing the quality assessment into content-/semantic-aware and distortion-aware aspects is actually quite common in IQA/VQA on 2D visual data. It’d be better to review some related QA metrics for 2D data, and analyze the key difference on the implementation of this idea between point-cloud data and 2D data quality assessment.
2. Lack of comparison on computational complexity.

**Questions:**

Please see the Weaknesses.

**Limitations:**

Please see the Weaknesses.

---

> ### Author Rebuttal · Authors · 2024-08-05
>
> We thank Reviewer #3 (hUdG) for the constructive comments. Our responses are as follows:
>
> **`R3-Weakness 1`**: Thank you for your insightful comments. We first review the related IQA/VQA papers and then analyze the key difference with our DisPA. The paper review will be added to our main paper.
>
> As for IQA, **CONTRIQUE** [1] learns distortion-related information on images with synthetic and realistic distortions based on contrastive learning. **Re-IQA** [2] trains two separate encoders to learn high-level content and low-level image quality features through an improved contrastive paradigm. **QPT** [3] also learns quality-aware representations through contrastive learning, where the patches from the same image are treated as positive samples, while the negative sample are categorized into content-wise and distortion-wise samples to contribute distinctly to the contrastive loss. **QPTv2** [4] is based on masked image modeling (MIM), which learns both quality-aware and aesthetics-aware representations through performing the MIM that considers degradation patterns.
>
> As for VQA, **CSPT** [5] learns useful feature representation by using distorted video samples not only to formulate content-aware distorted instance contrasting but also to constitute an extra self-supervision signal for the distortion prediction task. **DisCoVQA** [6] models both temporal distortions and content-related temporal quality attention via transformer-based architectures. **Ada-DQA** [7] considers video distribution diversity and employ diverse pretrained models to benefit quality representation. **DOVER** [8]  divides and conquers aesthetic-related and technical-related (distortion-related) perspectives in videos, introduces inductive biases for each perspective, including specific inputs, regularization strategies, and pretraining.
>
> Our DisPA's key difference with these papers lies in the utilization of unique characteristics of point clouds. For example, to fully investigate the quality information of point clouds, we **randomly rotate point clouds before projection** . In addition, the **mini-patch map integrates the patches from multi-view images** to take advantage of the multi-view characteristic of point clouds. Furthermore, it is worthwhile to note that our paper is first to use mutual information (MI) to achieve representation disentanglement, which has not been explored in IQA/VQA.
>
> **`R3-Weakness 2`**: Thank you for your comments. As your request, we compare the computational complexity of NR-PCQA models as follows:
>
> | Method       | Parameters (M) | Inference Time (s) |
> | ------------ | -------------- | ------------------ |
> | IT-PCQA      | 0.61           | 2.87               |
> | ResSCNN      | 1.23           | 1.92               |
> | PQA-Net      | 0.22           | 4.86               |
> | MM-PCQA      | 52.96          | 2.64               |
> | DisPA (ours) | 140.77         | 1.89               |
>
> From the above table, we can see that compared to the IT-PCQA, ResSCNN and PQA-Net whose architecture is specifically designed, our DisPA has large parameters because Vision Transformer (ViT) and Swin Transformer have been used, like MM-PCQA using ResNet-50. Instead of pretraining self-designed light-weight networks, we use popular encoders to make the pretrained encoders easier to be transferred to other tasks.
>
> However, although our DisPA has a larger model size, the inference time is the fastest, even including the rendering process, which demonstrates the efficiency of our DisPA.

---

> > ### Comment · Reviewer_hUdG · 2024-08-14
> >
> > Thank you for the rebuttal. The authors have effectively addressed my concerns. I will raise my score to 6.

---

### Official Review · Reviewer_oeD7 · 2024-07-12

**Soundness:** 3
**Presentation:** 3
**Contribution:** 3
**Rating:** 7
**Confidence:** 5

**Summary:**

This paper proposes a novel disentangled representation learning framework called DisPA to decouple the representation learning process of point cloud content and distortion. To sufficiently disentangle these two representations, the DisPA uses two branches to learn them and adopt different training philosophies separately. For the content-aware branch, DisPA pretrains one encoder using a proposed masked auto-encoding strategy, which partially masks the images projected from distorted point clouds and reconstructs the corresponding patches of the images projected from pristine point clouds; For the distortion-aware branch, the DisPA integrates the mini patches of the rendered multi-view images into a mini-patch map, which can focus on local distortions and ignore the global point cloud content. Furthermore, to disentangle the learned representations, the DisPA uses a trainable mutual information estimator to estimate the mutual information (actually the tight upper bound) between these two branches and further minimize it alternatively along with the training of the main network (i.e., the two encoders and regression layers). Finally, the experimental results demonstrate the superior performance of DisPA in terms of both prediction accuracy and generalizability.

**Strengths:**

+ This is the first paper exploring disentangled representation learning for PCQA. The disentanglement is reasonable and even necessary because the point cloud content and distortion are differently perceived by humans.

+ The proposed methodology is well-motivated. The masked auto-encoding strategy can intuitively make the encoder capture the point cloud content information.

+ The paper is written well. The motivation of why disentangled representation learning is essential for PCQA has been introduced clearly from the observation of human vision systems.

+ The mathematical derivations are detailed and easy to follow, including the approximation of MI in Section 3 and further proof in Appendix A.

**Weaknesses:**

1. From my viewpoint, the DisPA can be totally used for IQA, or even more suitable, since this work does not process 3D native point cloud data but just projects point clouds into images and uses 2D networks. Have the authors tried to introduce unique attribute information that is closely related to point clouds?
2. The whole DisPA is based on projections of point clouds, so the number of viewpoints is very important to the quality score prediction. However, the authors did not discuss the impact of the number of viewpoints.
3. How much time does it take to pretrain the content-aware encoder? The authors did not discuss the pretraining details in the implementation details part.
4. Why does not the LS-PCQA follow the K-fold data splitting? The authors should explain this for the loss of a unified experiment setting.
5. What is the actual architecture of the MI estimator? And what is the relation between the MI estimator and the lightweight neural network Q_phi? The idea of alternative training of the MI estimator and the main network is easy to understand, but the components of the MI estimator need more clarification. Is it just simple MLPs?
6. The mini-patch map generation has been used in many papers [1,2,3]. However, considering this is a minor contribution and the mini-patch map is actually effective for learning disentangled representations, this limited contribution is acceptable.
7. A minor weakness: There are some small flaws (gray bounding box) with the presentation of Figure 5 when zooming in. Please ensure all figures can be presented clearly without error.

[1] Wu, Haoning, et al. "Exploring video quality assessment on user generated contents from aesthetic and technical perspectives." Proceedings of the IEEE/CVF International Conference on Computer Vision. 2023.

[2] Wu, Haoning, et al. "Fast-vqa: Efficient end-to-end video quality assessment with fragment sampling." European conference on computer vision. Cham: Springer Nature Switzerland, 2022.

[3] Zhang, Zicheng, et al. "Gms-3dqa: Projection-based grid mini-patch sampling for 3d model quality assessment." ACM Transactions on Multimedia Computing, Communications and Applications 20.6 (2024): 1-19.

**Questions:**

1. Why the conditional distribution p(y|x) is unavailable in your case? This needs more explanations, because the mathematical analysis is somewhat abstract so more details are needed to make it more understandable.
2. What is the advantanges of the differential ranking loss function?
3. Can this work be trained end-to-end for IQA or video quality assessment (VQA) without modification of specific modules (just replacing the input/output)?
4. Can the masked auto-encoding strategy use natural images for pretraining?

**Limitations:**

Yes, the authors have adequately addressed the limitations and discussed the potential negative societal impact in the appendix D Limitations and Future Work.

---

> ### Author Rebuttal · Authors · 2024-08-05
>
> We thank Reviewer #2 (oeD7) for the insightful comments. Our responses are as follows:
>
> **`R2-Weakness 1`**: Thanks for your constructive comment. As you said, the DisPA can be used for IQA without changing the architecture and training pipeline. We use projected images instead of 3D native point cloud for the following reasons: **(1) Large scale pretraining.** In the scenario of PCQA, we focus on dense point clouds with huge data volume. Rendering point clouds into multi-view images can effectively reduce the data volume to facilitate large scale content-aware pretraining (i.e., the masked autoencoding strategy) with large batch size . **(2) Pixel-to-pixel correspondence.** After projection, the pixel-to-pixel correspondence can be established between the projected images of distorted and reference point clouds, which facilitates the computation of reconstruction loss for the content-aware pretraining. **(3) Generation of mini-patch map.** Projecting point clouds into images can enable the generation of mini-patch map for the distortion-aware branch to enhance the presentation of low-level distortion patterns.
>
> Furthermore, we have introduced some unique characteristics of point clouds to achieve better performance. For example, we **randomly rotate point clouds before projection** to fully investigate the quality information of point clouds. In addition, the **mini-patch map integrates the patches from multi-view images** to take advantage of the multi-view characteristic of point clouds.
>
> **`R2-Weakness 2`**: Thanks for your insightful comment and we have tested the performance of DisPA trained on SJTU-PCQA and WPC with different numbers of viewpoints (i.e., 2, 4, 6 and 12 views). The configuration of mini-patch map generation is accordingly modified. The viewpoints are all evenly distributed in the 3D space.
>
> | Performance | SJTU-PCQA |         |         |          | WPC     |         |         |          |
> | :---------- | :-------- | :------ | :------ | :------- | ------- | ------- | ------- | -------- |
> |             | 2 views   | 4 views | 6 views | 12 views | 2 views | 4 views | 6 views | 12 views |
> | SROCC       | 0.657     | 0.886   | 0.908   | 0.910    | 0.554   | 0.705   | 0.788   | 0.786    |
> | PLCC        | 0.636     | 0.893   | 0.919   | 0.919    | 0.577   | 0.711   | 0.790   | 0.793    |
>
> From the above table we can see that with the number of viewpoints increasing, the model performance first increases and then keep stabilized. More specifically, the performance with 12 viewpoints is slightly better than 6 viewpoints, and the performances of 2 and 4 viewpoints are relatively unsatisfactory. Considering the efficiency, we select 6 viewpoints in our paper. We will add these results in the revised paper.
>
> **`R2-Weakness 3`**: Thank you for your question. It takes about 20 hours for the content-aware pretraining on a single NVIDIA 3090 GPU because of the large dataset scale of LS-PCQA and the random rotations before projection during pretraining. We will add these details in our implementation part.
>
> **`R2-Weakness 4`**: We select K-fold because of the small dataset scale of SJTU-PCQA and WPC. However, LS-PCQA has **a large scale** so we select the train-val-test splitting. To further validate the stable performance on LS-PCQA, we test the the performance of several NR-PCQA models on LS-PCQA following unified 5-fold cross validation:
>
> | Performance | PQA-Net | IT-PCQA | GPA-Net | ResSCNN | MM-PCQA | DisPA(ours) |
> | ----------- | ------- | ------- | ------- | ------- | ------- | ----------- |
> | SROCC       | 0.583   | 0.337   | 0.587   | 0.593   | 0.587   | 0.623       |
> | PLCC        | 0.590   | 0.348   | 0.606   | 0.625   | 0.603   | 0.635       |
> | RMSE        | 0.199   | 0.226   | 0.192   | 0.170   | 0.191   | 0.161       |
>
> From the above table we can find the performances of NR-PCQA are relatively stable on LS-PCQA (compared with Table 1 of our main paper).
>
> **`R2-Weakness 5`**: As you said, the MI estimator is based on MLPs, as well as the $Q_\phi$. The $Q_\phi$ infers the variational distribution and the MI estimator computes the MI following Equation (3) in the main paper. Therefore, the **lightweight MLP $Q_\phi$ is the key component of MI estimator**, and MI estimator integrates the outputs of $Q_\phi$ and computes the MI.
>
> **`R2-Weakness 6`**: Thanks for your insightful comment. Although grid mini-patch sampling has been employed in the papers you listed, this is our sub-contribution and quite useful in our scenario when learning distortion-aware representations. Furthermore, our mini-patch sampling is different because we utilize the mutli-view characteristic of point clouds and integrates multi-view patches into the mini-patch map.
>
> **`R2-Weakness 7`**: Thanks for your kind comments. This is due to the issues of pdf browsers, we will fix this problem of image presentation.
>
> **`R2-Question 1`**: Thank you for your insightful question. In our case, the conditional distribution $p(\mathbf y |\mathbf x)$  denotes the distribution of distortion-aware representation given the content-aware representation. This is unavailable because the distribution of distortion patterns is unknown, since there is no prior knowledge of distortion types and intensities for DisPA during inference. Furthermore, there are cases of unseen distortion types for cross-dataset evaluations.
>
> **`R2-Question 2`**: The differential ranking loss can better assist the model to distinguish the quality difference when the point clouds have close quality labels.
>
> **`R2-Question 3`**: The DisPA can be trained end-to-end for IQA and VQA. However, the module of mini-patch map generation should be accordingly modified. For example, the IQA can be supposed as a special case of PCQA with only one viewpoint.
>
> **`R2-Question 4`**: The masked-encoding can definitely used on natural images. Acutually, the content-aware encoder is initialized with parameters optimized on ImageNet-1K with natural images.

---

> > ### Comment · Reviewer_oeD7 · 2024-08-09
> >
> > Thank you for the rebuttal. The authors have effectively addressed my concerns. I will raise my score to 7.

---

### Official Review · Reviewer_2ATd · 2024-07-13

**Soundness:** 3
**Presentation:** 2
**Contribution:** 2
**Rating:** 6
**Confidence:** 3

**Summary:**

This article's motivation is interesting. It combines content-aware and distortion-aware characteristics to train a 3D quality assessment network. Additionally, it employs a MAE-based method to train a content-aware encoder, uses patches to focus the network on learning distortion, and applies an MI module to integrate both features.

**Strengths:**

1. The authors first analyze the shortcomings of existing networks, specifically that data imbalance leads to overfitting in current methods, resulting in poor processing of other content images with the same degradation. Therefore, the authors aim to use an MAE-based method to learn content-related features.
2. The proposed key MI-based regularization is effective.
3. The presented methods can obtain impressive results on multiple datasets.

**Weaknesses:**

1. Regarding the masked part, there is a gap between the first and second stages because the input in the first stage is a partially masked image, while the input in the second stage is the complete image.
2. I am also a bit confused about why the constraint in the first stage is a clean/reference image. This would give the masked encoder the characteristic of restoration, while the core of quality assessment is to evaluate the quality of the image. If the image features are restored, will it affect the accuracy of the image quality assessment?
3. The motivation for the MI part should provide more details and explanations, which leaves me somewhat puzzled.
4. I am not quite sure what the distortion-aware encoder has learned. Is it truly related to degradation features? This might require some verification.

**Questions:**

This section corresponds to the Weaknesses:

1. How does the network address the gap between the input images in the first and second stages?
2. Why is a clean image used as a constraint? This results in learning the restoration characteristics, which is not very helpful for quality assessment.
3. The motivation for MI needs further explanation.
4. Visualize or analyze the distortion-aware features.

**Limitations:**

No potential negative societal impact. Please address the questions/suggestions above.

---

> ### Author Rebuttal · Authors · 2024-08-05
>
> We thank Reviewer #1 (2ATd) for the insightful comments. Our responses are as follows:
>
> **`R1-Weakness 1`**: Thank you for your in-depth comments. The gap between the partially masked image and the complete image **is addressed by fine-tuning**, as well as the main training objective (see Equation 10 in our paper). Our content-aware pretraining is based on masked auto-encoding (MAE), where the **MAE is also conducted on partially masked images and fine-tuning is conducted on complete images.**  The large-scale pretraining can enable the encoder to learn useful semantic representations, while the gap between masked and complete images can be addressed by adequately training the pretrained encoder with small-scale complete images.
>
> In our case, our MAE-based pretraining can make the encoder learn to recognize different point cloud contents (on LS-PCQA with a large scale of different point clouds), and the gap can be addressed fine-tuning the encoder on specific PCQA datasets. Furthermore, the pretraining can effectively enhance the prediction accuracy (see Table 3 in our paper) by a large margin, which indicates the gap has been sufficiently addressed after fine-tuning.
>
> **`R1-Weakness 2`**: Thanks for your comments. As you said, using a clean image as a constraint resembles image restoration. However, Since our goal is to learn disentangled representations, **we use a clean image as a constraint to force the content-aware encoder to ignore the distortions and capture point cloud content information from distorted projected images**. In contrast, if we use a distorted image as a constraint, the encoder tries to restore the images with distortion and will certainly learn joint features of distortion and content, which disobeys our goal of representation disentanglement.
>
> Furthermore, we use t-SNE visualization to verify that the learned content-aware representations are strongly correlated to point cloud content but not to distortions.  The visualization is in the `attached file Figure A`. The embeddings of different point cloud contents are separated by a clear boundary.
>
> **`R1-Weakness 3`**: Sorry for the lack of detailed explanation. **The mutual information (MI) part is designed to explicitly disentangle the representations of the two branches**, since the MI directly describes the correlation between two variables. In terms of detailed motivation of MI estimation and minimization (Section 3 in our paper), we have several key motivations:
>
> * Instead of computing the intractable exact value of MI (denoted as $\mathcal I$), we compute its upper bound $\hat{\mathcal I}$ because **minimizing the upper bound can also minimize the actual MI** as long as $\hat{\mathcal I}$ is a tight upper bound (i.e., close to the exact value of MI). Proof of the tight upper bound is in appendix A.
> * Since **we have no prior knowledge of the conditional distribution** $p(y|x)$ (reasons in R2-Question 1), **we can only approximate it with a variational distribution** $\hat{\mathcal I_v}$. Specifically, we use an MLP $\mathcal Q_\phi$ to infer a distribution $q_\phi(y|x)$ that is close to $p(y|x)$, and the inferred distributions of samples in a mini-batch are summarized to predict $\hat{\mathcal{I}}_v$ (see Equation 3 in our paper).
> * As mentioned above, minimizing the upper bound can only work when the upper bound is tight. **To make $\hat{\mathcal I_v}$ close to the tight upper bound $\hat{\mathcal I}$, we minimize the KL divergency between $p(y|x)$ and $q_\phi(y|x)$**. After simplification, the KL divergency can formulated as the $\mathcal L_\text{MI}$​ (Equation 5 in our paper).
> * Finaly, for each epoch during the training, we first train the MLP $\mathcal Q_\phi$ (i.e., minimizing the KL divergency between$p(y|x)$ and $q_\phi(y|x)$). Then we use the trained $\mathcal Q_\phi$ to estimate the MI to regularize the encoders, as shown in Equation 10 in our paper, where the estimated $\hat I_v$ is used as a regularization term to achieve explicit disentanglement.
>
> **`R1-Weakness 4`**: To verify that the distortion-aware features are related to degradations, we conduct the t-SNE visualization of learned representations of the distortion-aware branch. The results are in `attached file Figure A`. We also present the visualization results of PQA-Net and GPA-Net for comparison, because these two methods both use distortion type prediction to learn distortion-aware representations. From `attached file Figure A` we can see that **the learned distortion-aware representation shows clear and separate clustering for different distortion types, indicating a strong correlation with degradations.** In addition, the content-aware representations present worse clustering for distortion types.
>
> In addition, many papers [1,2,3] have verified that the mini-patch map can force the network to learn features that are related to degradations.
>
> [1] Fast-vqa: Efficient end-to-end video quality assessment with fragment sampling
>
> [2] Exploring video quality assessment on user generated contents from aesthetic and technical perspectives
>
> [3] Gms-3dqa: Projection-based grid mini-patch sampling for 3d model quality assessment

---

> > ### Comment · Reviewer_2ATd · 2024-08-09
> > **Response to the authors.**
> >
> > Thank you for your detailed responses. I apologize for missing the fine-tuning process in the first question. In fact, applying the fine-tuned operation for MAE is general, and I thought this step was omitted in the paper, which caused my confusion. Other doubts have also been resolved, and I will improve my score to weak accept.

---

### Author Rebuttal · Authors · 2024-08-05

Dear Reviewers and ACs:

We sincerely appreciate all the reviews. They give insightful and high-quality comments on our paper. We would like to emphasize that our work proposes a novel and effective disentangled representation learning framework for point cloud quality assessment called DisPA. We appreciate your recognition of our efforts in attempting to disentangle the content-aware and distortion-aware representations via mutual information minimization.

We have carefully read your comments and made responses to all the reviewers, where we have supplemented our work with additional experiments and visualization results to incorporate the insightful suggestions of the reviewers. **The related figures are attached in the pdf file.** Thank you all for the valuable suggestions.

Thanks,

Paper 6653 Authors.

---

### Decision · Program_Chairs · 2024-09-25

**Decision:**

Accept (poster)

**Comment:**

Following the author-reviewer discussion stage, three reviewers expressed a positive opinion of the paper. While reviewer nBew raised concerns regarding the presentation, no critical technical issues were mentioned. The AC reviewed the authors' response to reviewer nBew and believes that these concerns can be addressed in the preparation of the camera-ready version. Therefore, the AC recommends acceptance.